# Integrating Physical and Biochemical Cues for Muscle Engineering: Scaffolds and Graft Durability

**DOI:** 10.3390/bioengineering11121245

**Published:** 2024-12-09

**Authors:** Farbod Yousefi, Lauren Ann Foster, Omar A. Selim, Chunfeng Zhao

**Affiliations:** 1Department of Orthopedic Surgery, Mayo Clinic, Rochester, MN 55905, USA; yousefi.farbod@mayo.edu (F.Y.); lauren.foster2@emory.edu (L.A.F.); selim.omar@mayo.edu (O.A.S.); 2Atlanta Veterans Affairs Medical Center, Emory University School of Medicine, Atlanta, GA 30307, USA

**Keywords:** skeletal muscle regeneration, extracellular matrix, mechanotransduction, external cues, cellular senescence, fibrosis, tissue scaffolds, pharmacotherapy, exosomes

## Abstract

Muscle stem cells (MuSCs) are essential for skeletal muscle regeneration, influenced by a complex interplay of mechanical, biochemical, and molecular cues. Properties of the extracellular matrix (ECM) such as stiffness and alignment guide stem cell fate through mechanosensitive pathways, where forces like shear stress translate into biochemical signals, affecting cell behavior. Aging introduces senescence which disrupts the MuSC niche, leading to reduced regenerative capacity via epigenetic alterations and metabolic shifts. Transplantation further challenges MuSC viability, often resulting in fibrosis driven by dysregulated fibro-adipogenic progenitors (FAPs). Addressing these issues, scaffold designs integrated with pharmacotherapy emulate ECM environments, providing cues that enhance graft functionality and endurance. These scaffolds facilitate the synergy between mechanotransduction and intracellular signaling, optimizing MuSC proliferation and differentiation. Innovations utilizing human pluripotent stem cell-derived myogenic progenitors and exosome-mediated delivery exploit bioactive properties for targeted repair. Additionally, 3D-printed and electrospun scaffolds with adjustable biomechanical traits tackle scalability in treating volumetric muscle loss. Advanced techniques like single-cell RNA sequencing and high-resolution imaging unravel muscle repair mechanisms, offering precise mapping of cellular interactions. Collectively, this interdisciplinary approach fortifies tissue graft durability and MuSC maintenance, propelling therapeutic strategies for muscle injuries and degenerative diseases.

## 1. Introduction

Muscle stem cells (MuSCs) are central to skeletal muscle regeneration, playing an indispensable role in maintaining the tissue’s structural integrity and function. These cells are responsive to an intricate network of mechanical, biochemical, and molecular signals [1,2]. The regenerative capacity of skeletal muscle hinges on MuSCs’ ability to sense and adapt to their microenvironment, ensuring efficient repair and sustained functionality [3].

Biomaterial scaffolds have emerged as pivotal tools in enhancing MuSC function and muscle regeneration. These scaffolds not only provide structural support but also mimic the native extracellular matrix (ECM), facilitating cell adhesion, proliferation, and differentiation [4]. Various biomaterials, such as natural polymers like collagen and synthetic polymers like polylactic acid (PLA), have been utilized to create scaffolds that promote MuSC attachment and growth [5]. Moreover, these scaffolds can be functionalized with bioactive molecules to enhance myogenic differentiation and tissue regeneration [6].

Mechanosensitive pathways are critical in regulating MuSC behavior, where mechanical forces like shear stress, tensile stress, and cyclic stretching are converted into biochemical signals that influence key cellular processes such as migration, differentiation, and ECM organization [7,8]. These mechanical cues are integrated by the cytoskeleton, which plays a vital role in cellular adaptation and functionality. Biomaterial scaffolds can be engineered to present specific mechanical properties and topographical cues that modulate mechanotransduction pathways in MuSCs [9]. For instance, scaffolds with aligned nanofibers have been shown to direct MuSC alignment and promote myogenic differentiation [10]. Additionally, meso-scale topological cues activate mechanotransduction pathways, particularly through focal adhesion kinase (FAK) and cytoskeletal proteins, driving MuSC activation and enhancing ECM production [11].

Epigenetic modifications tightly regulate MuSC states, particularly during quiescence and activation. The hypermethylation of genes involved in cell cycle progression and differentiation maintains MuSC dormancy, while demethylation upon injury triggers their activation [12]. Histone modifications also play a crucial role, with acetylation facilitating the expression of myogenic regulatory factors like MyoD, while deacetylation by HDACs leads to gene repression, maintaining quiescence [13]. The balance of histone methylation marks, such as H3K4me3 (promoting transcription) and H3K27me3 (repressing genes), is crucial for proper MuSC function during regeneration [14]. Moreover, chromatin remodelers like the SWI/SNF complex adjust DNA accessibility, further regulating MuSC activation [15]. Metabolic pathways also influence MuSC behavior, with glutamine playing a key role in supporting the TCA cycle, enhancing both self-renewal and regenerative potential [16].

Aging significantly impairs MuSC function, characterized by a reduction in MuSC numbers and a decline in their self-renewal capacity. This decline severely limits the muscle’s ability to repair and regenerate post-injury [17]. Cellular senescence further exacerbates this problem, reducing the MuSC pool and impairing muscle repair mechanisms [18]. The aged MuSC niche undergoes detrimental changes, including dysfunctional fibro/adipogenic progenitors (FAPs), increased fibrosis, and a rise in adipogenesis [19]. Aged MuSCs also exhibit altered metabolic activity, shifting from glycolysis to a greater reliance on oxidative phosphorylation, which leads to increased reactive oxygen species (ROS) levels and oxidative damage [20]. This accumulation of ROS not only reduces the regenerative capacity of MuSCs but also heightens their susceptibility to apoptosis. Ref. [21] explores how excessive ROS induces apoptosis in fibroblasts by mitochondrial dysfunction, which parallels how ROS accumulation in MuSCs can reduce their regenerative capacity and increase apoptosis susceptibility. Ref. [22] discusses how high glucose levels trigger ROS generation, leading to mitochondrial dysfunction and apoptosis, relevant to the impact of ROS on MuSCs.

To mitigate the detrimental effects of ROS on MuSCs, biomaterials with antioxidant properties have been developed. For example, scaffolds incorporating antioxidant molecules like melatonin can scavenge excess ROS, thereby enhancing MuSC survival and function [23]. Additionally, hydrogels loaded with ROS-scavenging nanoparticles can modulate the oxidative environment, promoting muscle regeneration in aged tissues [24]. Conversely, controlled ROS generation from biomaterials can be utilized to trigger signaling pathways that promote MuSC activation and differentiation [25,26].

Pax7 is a paired box transcription factor that serves as a pivotal regulator of muscle stem cells (MuSCs). Highly expressed in quiescent MuSCs, Pax7 is essential for maintaining the myogenic lineage and self-renewal capacity of these cells [27]. It functions by activating target genes that promote MuSC survival and by repressing differentiation-inducing factors, thus preserving the stem cell pool required for effective muscle regeneration [28]. Research indicates that Pax7 expression levels fluctuate during MuSC activation and differentiation, with elevated levels sustaining quiescence and self-renewal, while downregulation corresponds with commitment to myogenic differentiation [29]. Understanding the regulation of Pax7 is critical for developing strategies to manipulate MuSC fate decisions, offering potential therapeutic avenues for muscle repair and regeneration.

Several signaling pathways are integral to MuSC regulation, controlling their quiescence, activation, and differentiation (Figure 1). The Notch pathway is pivotal in maintaining quiescence by preventing premature differentiation and the expression of myogenic factors like MyoD [30]. Wnt signaling, particularly in its non-canonical form, can inhibit myogenic differentiation, highlighting its complex role in MuSC regulation [31]. The Hedgehog signaling pathway supports MuSC activation and differentiation into myogenic lineages, with disruptions potentially leading to impaired regeneration and increased fibrosis [32]. Furthermore, the PI3K/Akt pathway, activated by insulin-like growth factor 1 (IGF-1), promotes MuSC proliferation and differentiation, with IGF-1 delivery enhancing muscle regeneration and recovery [33]. METTL3, through the regulation of the Notch pathway, further supports stemness and maintains MuSC states [34].

## 2. Challenges in Muscle Regeneration

### 2.1. Volumetric Muscle Loss (VML)

Volumetric muscle loss (VML) presents significant challenges in muscle regeneration, primarily due to the extensive damage it causes to both muscle tissue and its microenvironment. Agent-based modeling has underscored the critical role of the MuSC microenvironment in failed regeneration, suggesting that an in-depth understanding of these environmental dynamics is essential for developing effective treatments [36]. VML injuries trigger a complex inflammatory response, which initially aids in recruiting and activating progenitor cells. However, when this inflammation becomes chronic, it can hinder key regenerative processes like myofiber bridging and myotendinous junction formation, thereby limiting overall muscle regeneration [37]. Studies using murine models have shown that VML defects in muscles like the tibialis anterior can lead to significant motoneuron axotomy and a reduction in neuromuscular junction (NMJ) formation, outcomes that are closely linked to the depletion of MuSCs [38]. Current therapeutic strategies for VML involve both cellular and acellular approaches aimed at restoring lost tissue and function, with decellularized ECM scaffolds offering a promising option due to their ability to support host cell infiltration and tissue regeneration, along with advantages like scalability and cost-effectiveness [39,40].

### 2.2. Fibrosis and Scar Formation

Fibrosis and scar formation are major obstacles in the effective regeneration of muscle tissue following VML injuries. Fibroblast-mediated collagen deposition, while essential for initial wound stabilization, often results in the formation of non-functional scar tissue that impedes muscle regeneration [41]. Collagen I, primarily deposited by fibroblasts, dominates the healing response and hampers the restoration of functional muscle tissue [42]. The challenge lies in balancing fibroblast activity and the inflammatory response to promote functional muscle repair while minimizing fibrosis. Innovative scaffold designs and therapeutic strategies are being explored to achieve this balance, aiming to reduce scar tissue formation and enhance the overall quality of muscle regeneration [43]. However, fibroblast-mediated scar formation remains a significant complication, creating a fibrotic environment that further hinders effective muscle repair [40].

### 2.3. Immune Response and Inflammation

The immune response plays a dual role in muscle regeneration, initially supporting tissue repair but potentially leading to chronic inflammation if not properly regulated. Chronic inflammation, particularly in the context of aging, disrupts the MuSC niche, resulting in impaired MuSC function and diminished regenerative capacity [44]. During the regeneration process, the immune response must transition from a pro-inflammatory to an anti-inflammatory phase. This involves macrophages shifting from an M1 phenotype, which is pro-inflammatory, to an M2 phenotype, which supports tissue repair. Regulatory T cells (Tregs) also play a crucial role in this phase [45]. Fibro/adipogenic progenitors (FAPs) contribute to muscle regeneration by depositing ECM collagen, which provides the structural support necessary for the alignment and fusion of myogenic cells (Figure 2) [46]. Anti-inflammatory drugs are being explored as a means to promote a pro-regenerative environment in VML injuries by reducing chronic inflammation, which otherwise inhibits critical processes like myofiber bridging [37].

### 2.4. Cellular Senescence and Its Impact

Cellular senescence poses a significant challenge in muscle regeneration, particularly in the context of aging and chronic injury. Senescent cells secrete pro-inflammatory cytokines and matrix-degrading enzymes that exacerbate tissue damage and inhibit regeneration, contributing to a detrimental environment for muscle repair [49]. Therapeutic strategies targeting senescent cells, known as senolytics, show promise in enhancing muscle regeneration. For instance, Dasatinib, a tyrosine kinase inhibitor, selectively induces apoptosis in senescent cells and, when combined with Quercetin, an anti-inflammatory flavonoid, improves muscle function and regeneration in aged mice [50]. Another promising senolytic agent is Navitoclax, which targets BCL-2 family proteins involved in cell survival. However, its clinical use is limited due to potential side effects such as thrombocytopenia [37]. Modulating the immune environment by promoting macrophage polarization towards the M2 phenotype through cytokines like IL-4 and IL-13 is another strategy being investigated, as it can create a more favorable environment for muscle regeneration [46].

## 3. Microenvironment and Extracellular Matrix (ECM)

The microenvironment, or niche, refers to the local cellular environment surrounding MuSCs, encompassing not only the extracellular matrix (ECM) but also neighboring cells, soluble factors, and physical conditions [51]. The ECM is a vital component of this microenvironment, comprising a complex network of proteins and polysaccharides that provide structural support and biochemical cues to cells [52]. While the ECM focuses on the non-cellular structural components, the microenvironment includes all elements that influence MuSC behavior, such as growth factors, cytokines, cell–cell interactions, and mechanical stimuli [53]. The key difference between the microenvironment and the ECM lies in their scope: the ECM is a structural and biochemical scaffold, whereas the microenvironment is a broader concept that integrates the ECM with other factors affecting stem cell fate (Figure 3).

### 3.1. ECM’s Role in Muscle Regeneration

The extracellular matrix (ECM) plays a critical role in muscle regeneration, particularly in the context of volumetric muscle loss (VML) injuries. These injuries severely compromise or destroy the basement membrane, which is essential for guiding regenerative processes [54]. When the ECM is lost in such injuries, the ability of muscle stem cells (MuSCs) to migrate and fuse into myotubes is significantly impaired. This results in MuSCs migrating laterally outside the disrupted basement membrane, which hinders effective muscle repair [55]. The MuSC microenvironment, or niche, is vital for maintaining stemness, with ECM components and cytokines providing the necessary support for MuSC maintenance and activation [56]. Thus, muscle regeneration is a complex process influenced by the interplay between the MuSC microenvironment, signaling pathways, and ECM components [57].

### 3.2. Mechanical and Biochemical Cues in the ECM

Mechanical and biochemical cues within the ECM are fundamental in regulating stem cell behavior. Mechanical forces are key to tissue patterning, converting physical cues from the environment into biochemical signals that regulate stem cell migration, differentiation, and ECM organization [58]. The ECM not only provides structural support but also delivers biochemical signals that guide stem cell fate. The cytoskeleton integrates these signals, ensuring proper cellular adaptation and functionality [59]. Focal adhesions serve as the connection between the ECM and the cytoskeleton, transmitting mechanical signals that influence gene expression and cellular responses [60]. Additionally, surface features at the microscale and nanoscale can guide cell behavior by affecting cytoskeletal alignment and focal adhesion dynamics, which in turn influence stem cell differentiation and tissue integration [61].

### 3.3. Influence of ECM Stiffness and Structure

The stiffness and structural composition of the ECM significantly influence MuSC fate and muscle regeneration. Glycoproteins, proteoglycans, and collagens within the ECM provide crucial structural support and regulate the availability of growth factors that impact MuSC behavior [62]. Fibronectin and laminins, for example, are essential for maintaining tissue integrity and repair. Fibronectin, in particular, enhances WNT7a activity, which promotes MuSC proliferation and self-renewal [63]. In 3D culture systems, Type I collagen supports myogenic progression, while Type V and VI collagens interact with specific receptors to maintain MuSC quiescence or delay differentiation [64]. The need for scaffolds that retain the 3D architecture of the native ECM is paramount for mimicking the properties of muscle tissue. However, decellularized ECM scaffolds often present challenges due to the unknown specifics of their composition [65].

### 3.4. Challenges in ECM Restoration

Restoring the ECM in cases of VML injuries presents considerable challenges. The extensive damage to both muscle tissue and the ECM leads to impaired regenerative processes, making effective restoration difficult [65]. The loss of the basement membrane in these injuries disrupts the essential biophysical and biochemical cues required for proper MuSC behavior, resulting in compromised MuSC migration and myotube fusion [64]. Furthermore, fibroblast-mediated collagen deposition and the formation of scar tissue create a fibrotic environment that complicates muscle regeneration [62]. Although innovative therapeutic approaches, including biocompatible scaffolds and pharmacological agents, are being developed to restore the lost ECM environment, challenges remain in achieving long-term stability, functionality, and scalability (Table 1) [63].

**Table 1 bioengineering-11-01245-t001:** Studies on molecular pathways and mechanisms regulating MuSC states. This table summarizes key studies on molecular pathways and mechanisms regulating MuSC states, highlighting the significant proteins, actions, cellular responses, and outcomes involved in these processes.

Ref.	Key Proteins/Factors	Mechanistic Action	Cellular Response	Key Outcome
[66]	NUMB, Template DNA	Asymmetric division and cosegregation of template DNA	Maintenance of stem cell self-renewal	Preservation of stem cell pool
[67]	Template DNA	Asymmetric division in cancer stem cells	Maintenance of self-renewal capacity	Stem cell self-renewal
[68]	WNT7A	WNT signaling pathway activation	Expansion of self-renewing cells	Enhanced stem cell pool expansion
[69]	WNT signaling	Regulation of self-renewal and cardiac stem cells	Maintenance and expansion of cardiac stem cells	Regeneration of infarcted myocardium
[70]	JAK/STAT	Promotion of symmetric division	Expansion of self-renewing cell population	Increased stem cell pool
[71]	JAK/STAT	Maintenance of stem cell quiescence	Promotion of self-renewal	Preservation of stem cell population
[72]	RNAPII	Transcriptional regulation	Maintenance of quiescence	Prevention of premature activation
[73]	ATR, Cyclin F-SCF	Regulation of genome integrity and cell cycle factors	Preservation of quiescence	Maintenance of stem cell longevity
[74]	Cyclin F-SCF	Degradation of cell cycle proteins	Maintenance of quiescence	Stem cell preservation
[75]	PI3K/AKT	Rescues quiescence defects	Rejuvenation of aged MuSCs	Restoration of stem cell balance
[76]	Notch, miR-708, KLF7	Delay of cell cycle reentry	Maintenance of stem cell pool	Prevention of premature depletion
[77]	MuSK, BMP	Niche-specific signaling	Regulation of myofiber size	Maintenance of quiescence
[78]	N-cadherin, M-cadherin	Niche adhesion	Transition from quiescence to activation	Stem cell activation
[79]	Cadherins	Cadherin-dependent adhesion	Niche anchorage	Preservation of stem cell readiness
[80]	Collagen, Laminin	Structural support through ECM components	Maintenance of homeostasis	Support for tissue regeneration
[81]	ECM components	Structural support for tissue regeneration	Maintenance of tissue integrity	Enhanced regeneration capability
[82]	H3K27me3	Regulation of gene expression during myogenesis	Differentiation-dependent gene activation	Promotion of muscle differentiation
[83]	H3K4me2, H3K4me3	Marking of muscle-relevant genes during myogenesis	Promotion of gene expression	Enhanced muscle differentiation
[84]	Smad2, LEF1	Regulation of histone modifications during differentiation	Control of gene expression	Induction of differentiation
[85]	eIF2α	Regulation of gene activation	Maintenance of quiescence	Prevention of premature differentiation
[86]	H3K27me3	Maintenance of repressive chromatin state	Preservation of quiescence	Prevention of premature activation
[87]	Notch	Prevention of muscle atrophy	Maintenance of quiescence	Prevention of muscle atrophy

## 4. Scaffold Design and Engineering

Scaffold design in muscle tissue engineering is fundamentally guided by the need to replicate the native extracellular matrix (ECM) and provide a conducive environment for muscle regeneration [88]. The basic rationale involves creating scaffolds with physical properties that support MuSC adhesion, proliferation, differentiation, and organization into functional muscle tissue [89]. Key physical properties include mechanical strength, elasticity, porosity, and biodegradability, which must be carefully tuned to match those of native muscle tissue [90].

### 4.1. Scaffold Properties Mimicking ECM

Scaffolds that closely mimic the properties of the extracellular matrix (ECM) are vital in enhancing muscle regeneration. By enabling crosstalk between physical, biological, material, and pharmacological factors, these scaffolds optimize intracellular signaling pathways and cellular responses, thereby improving regenerative outcomes [91]. Integrating scaffold designs with pharmacotherapy presents a promising strategy to overcome transplantation challenges, offering mechanical and biochemical cues that enhance both the functionality and endurance of grafts [92]. Pharmacotherapy, such as growth factors and senolytic drugs, works synergistically with scaffold properties to modulate MuSC behavior, supporting cell survival, proliferation, and differentiation through engagement with key intracellular pathways [93]. Notably, scaffold alignment has been shown to promote proliferation by activating integrin-mediated signaling via the PI3K/Akt pathway [63].

Physical properties such as stiffness and elasticity are critical, as they influence cell behavior through mechanotransduction pathways [94]. Muscle tissue has a specific range of stiffness, and scaffolds must match this to promote proper MuSC differentiation and function [4]. Additionally, the scaffold’s porosity and pore size are important for nutrient diffusion, vascularization, and waste removal, which are essential for cell survival and tissue integration [95,96]. Biodegradability is another key property, allowing the scaffold to gradually degrade as new tissue forms, reducing the need for surgical removal (Figure 4) [97,98,99].

### 4.2. Integration of Mechanical Cues in Scaffolds

The mechanical forces exerted by scaffold structures play a crucial role in influencing cellular mechanotransduction, which in turn impacts gene expression and protein synthesis—key processes for maintaining MuSC stemness and enhancing the longevity of muscle grafts [100]. The dynamic interplay between mechanical loading and biochemical signaling within the scaffold environment is essential for promoting tissue integration and functional recovery [101]. Conductive nanofibers, for example, have been found to enhance myotube formation during differentiation by activating the MAPK/ERK pathway [102]. Additionally, scaffold stiffness is critical in regulating cell behavior, particularly through the activation of the RhoA/ROCK pathway during differentiation (Figure 5) [103].

Materials suitable for muscle tissue engineering include both natural and synthetic polymers. Natural polymers like collagen, gelatin, and fibrin offer excellent biocompatibility and bioactivity, closely resembling the native ECM [104]. However, they may have limited mechanical strength and variable degradation rates. Synthetic polymers such as polylactic acid (PLA), polyglycolic acid (PGA), and polycaprolactone (PCL) provide tunable mechanical properties and degradation rates but may lack cell recognition sites [105]. Combining natural and synthetic materials can leverage the advantages of both, creating composite scaffolds that meet the necessary physical and biological requirements [106].

### 4.3. Pharmacotherapy Interactions with Scaffolds

Pharmacological interventions aimed at enhancing scaffold durability and preserving muscle youthfulness and stemness are emerging as promising approaches in regenerative medicine [107]. The incorporation of bioactive molecules within scaffolds allows for targeted delivery and sustained release, ensuring the continuous stimulation of MuSCs and bolstering the regenerative capacity of muscle tissues [108]. Furthermore, manipulating immune cells via MSC exosomes within scaffolds can modulate macrophage activity, reducing inflammation and promoting muscle regeneration [109]. MSC-conditioned medium has also been shown to reduce inflammation and fibrosis while promoting the formation of new muscle fibers, particularly in immunocompetent mice [110].

### 4.4. Advances in Scaffold Manufacturing

Advancements in scaffold manufacturing techniques and material enhancements are critical for overcoming the challenges associated with scaling decellularized ECM scaffolds [111]. Three-dimensional printing technology, in particular, allows for the fabrication of customized scaffolds that can be tailored to match specific injury sites, offering personalized solutions for muscle regeneration [112]. Materials used in 3D printing, such as polylactic acid (PLA), polycaprolactone (PCL), and hydrogels, provide tunable mechanical properties and biocompatibility, making them ideal for muscle scaffold applications [102]. Combining 3D printing with electrospinning techniques can further enhance scaffold design by providing structural support and a microenvironment conducive to effective tissue regeneration [113].

### 4.5. Novel Materials for Scaffold Design

The development of novel materials for scaffold design focuses on achieving biodegradability and tunable mechanical properties, essential for long-term use in tissue regeneration. Materials such as PLA, PCL, and hydrogels are engineered to possess these qualities, offering the necessary stiffness and elasticity for effective integration [114]. Biomimetic hydrogels, with their high water content and tunable mechanical properties, mimic the physical and biochemical properties of the native ECM, supporting cell growth, proliferation, and differentiation [11]. Dynamic hydrogels, which respond to environmental stimuli like pH or mechanical forces, adapt to the needs of regenerating tissue, thereby enhancing tissue integration [115]. Additionally, hybrid scaffolds that combine hydrogels with electrospun fibers or 3D-printed frameworks provide both mechanical support and biological activity, optimizing the tissue repair process (Table 2) [116].

**Table 2 bioengineering-11-01245-t002:** Summary of scaffold properties in muscle tissue engineering. The table categorizes scaffolds by type, physiological properties, physicochemical properties, and composition.

Ref.	Scaffold Type	Physiological Properties	Physicochemical Properties	Composition
[88]	Poly(glycerol sebacate)-gelatin	Supports cell adhesion and proliferation; promotes muscle regeneration	Biodegradable; tunable mechanical strength and elasticity	Poly(glycerol sebacate), gelatin
[117]	Collagen-based scaffold with endothelial cells	Enhances vascularization; supports muscle tissue formation	Biocompatible; mimics native ECM stiffness	Collagen type I, endothelial cells
[98]	3D bioprinted scaffold	Customizable shape for defect site; promotes cell viability	Controlled porosity; mechanical strength suitable for muscle tissue	Polycaprolactone (PCL), bioink with cells
[105]	Synthetic polymer scaffold	Supports stem cell differentiation	Adjustable degradation rate; tunable mechanical properties	Polylactic acid (PLA), polyglycolic acid (PGA)
[4]	Elastic substrate scaffold	Regulates stem cell self-renewal and differentiation	Elastic modulus matching muscle tissue; biodegradable	Hydrogel substrates of varying stiffness
[55,89]	Injectable hydrogel scaffold	Facilitates cell delivery; supports tissue integration	Shear-thinning; self-healing properties	Hyaluronic acid, gelatin methacryloyl
[94]	Matrix elasticity-modulated scaffold	Directs stem cell lineage specification	Variable stiffness; elastic properties	Polyacrylamide gel with tunable crosslinking
[106]	Composite scaffold	Enhances mechanical strength; supports cell attachment	Biodegradable; controlled porosity	Collagen, hydroxyapatite
[95]	Electrospun nanofiber scaffold	Mimics native ECM structure; supports cell infiltration	High surface area; adjustable fiber diameter	Polycaprolactone (PCL) nanofibers
[118]	Decellularized ECM scaffold	Provides natural biochemical cues; supports regeneration	Preserved ECM architecture; natural mechanical properties	Decellularized muscle tissue ECM

## 5. Emerging Therapies and Technologies

### 5.1. Pluripotent Stem Cells and Myogenic Progenitors

Human pluripotent stem cells (hPSCs), which can be derived from human embryonic stem cells (hESCs) or induced pluripotent stem cells (iPSCs), hold significant promise for muscle regeneration. These cells can be directed to differentiate into myogenic progenitors using protocols that involve specific small molecules and growth factors, leading to the expression of key markers like Pax7 and MyoD [119]. The differentiation process mirrors embryonic myogenesis, where transcription factors such as Pax3, Pax7, MyoD, and Myf5 guide the cells toward a myogenic fate [102]. Once derived, these myogenic progenitors can be expanded in vitro and transplanted into injured muscle tissue, where they integrate, proliferate, and differentiate into mature muscle fibers [101]. A notable advantage of using iPSC-derived myogenic progenitors is the potential for autologous transplantation, which significantly reduces the risk of immune rejection [93]. However, scalability remains a major challenge, as producing sufficient quantities of myogenic progenitors for clinical applications requires advances in bioreactor technology and scalable culture systems [120].

### 5.2. Three-Dimensional Printing and Custom Scaffolds

Three-dimensional printing technology has revolutionized scaffold design by enabling precise control over scaffold architecture and the delivery of bioactive cues, making it ideal for creating scaffolds that closely mimic the complex structure of native muscle tissue [121]. By incorporating multiple cell types and bioactive molecules during the 3D printing process, complex tissue constructs that promote cell proliferation and tissue integration can be created [122]. Electrospinning, a complementary technique, produces nanofibrous scaffolds that resemble the ECM of native muscle tissue, offering a high surface area and porosity for cell attachment and nutrient exchange [123]. These electrospun scaffolds can be functionalized with bioactive molecules to enhance cell adhesion and differentiation, and their fiber alignment can effectively guide the regeneration of muscle fibers [124]. Additionally, methods like “Drydux Transduction” have been developed to increase activation efficiency during transduction with viral vectors, further enhancing the functionality of these custom scaffolds [125].

### 5.3. Nanotechnology and Biomimetic Materials

Nanotechnology is playing an increasingly critical role in developing biomimetic materials that closely replicate the properties of native ECM. Electrospun nanofibrous scaffolds, which mimic the ECM’s structure, offer a high surface area and porosity essential for cell attachment and nutrient exchange, thus supporting muscle tissue regeneration [123]. Biomimetic hydrogels, designed to imitate the physical and biochemical properties of native ECM, provide an optimal environment for cell growth, proliferation, and differentiation due to their high water content and tunable mechanical properties [126]. Dynamic hydrogels, which can respond to environmental stimuli such as pH or mechanical forces, further enhance tissue integration by adapting to the specific needs of regenerating tissues [127]. Additionally, nanotopographical cues delivered via engineered extracellular vesicles (EVs) have shown potential in enhancing proliferation and repairing aged muscle tissue through sequential administration [124]. The combination of 3D printing with electrospinning continues to refine scaffold design, providing both structural support and a conducive microenvironment for effective tissue regeneration [128].

## 6. Cellular Mechanisms and Metabolic Pathways

### 6.1. Cytoskeletal Dynamics in Mechanotransduction

The cytoskeleton, a dynamic network composed of actin filaments, microtubules, and intermediate filaments, plays a central role in mechanotransduction—the process by which cells convert mechanical signals into biochemical responses. Focal adhesions serve as crucial connectors between the ECM and the cytoskeleton, transmitting mechanical signals that influence gene expression and cellular behavior [129,130]. The stiffness of the ECM and substrate rigidity initiate signaling pathways such as YAP/TAZ, Rho/ROCK, and FAK, which are key to stem cell mechanotransduction [131]. The cytoskeleton’s unique mechanical properties, including entropic and enthalpic elasticity, allow it to stiffen nonlinearly under shear stress, enhancing cellular resistance to deformation and maintaining structural integrity under mechanical stress [132,133].

### 6.2. Metabolic Pathways in MuSC Function

Muscle stem cells (MuSCs) rely on distinct metabolic pathways to maintain their function and support muscle regeneration. In their quiescent state, MuSCs primarily utilize oxidative phosphorylation (OXPHOS) to sustain a low metabolic profile. However, upon activation, they shift to glycolysis to meet the increased energy demands required for cell division and biosynthetic activities [134]. Fatty acid oxidation (FAO) is another critical metabolic pathway, providing ATP and acetyl-CoA necessary for energy homeostasis and histone acetylation, with enhanced FAO linked to improved self-renewal and reduced differentiation of MuSCs [135]. Reactive oxygen species (ROS), generated as byproducts of cellular metabolism, serve as signaling molecules that promote MuSC activation at low levels but can cause oxidative stress and impair MuSC function if accumulated excessively [136]. The pentose phosphate pathway (PPP) is vital during MuSC activation, supplying ribose-5-phosphate for nucleotide synthesis and NADPH for reductive biosynthesis and antioxidant defense, supporting anabolic demands and protecting against oxidative stress [137]. Additionally, amino acid metabolism, especially involving branched-chain amino acids (BCAAs), regulates protein synthesis via the mTOR pathway, influencing MuSC growth and differentiation [138].

### 6.3. Epigenetic Regulation in Muscle Regeneration

Epigenetic mechanisms are crucial in regulating MuSC behavior and muscle regeneration. During quiescence, the hypermethylation of specific genes involved in cell cycle progression and differentiation maintains MuSC dormancy, while demethylation upon muscle injury activates these genes, triggering regeneration [139]. Histone modifications, such as the methylation of H3K4me3 (associated with active transcription) and H3K27me3 (associated with gene repression), play a significant role in regulating gene expression, with the balance between these modifications being crucial for MuSC regeneration [140]. MicroRNAs (miRNAs) like miR-1, miR-206, and miR-133 are also involved, fine-tuning gene expression by regulating mRNA degradation or translational repression, thus influencing MuSC proliferation and differentiation [141]. Long non-coding RNAs (lncRNAs) interact with chromatin-modifying complexes to modulate gene expression, further impacting MuSC function [13]. Chromatin remodelers, such as the SWI/SNF complex, are essential for MuSC activation as they reposition nucleosomes, altering DNA accessibility to transcription factors [142].

### 6.4. Mechanical Forces and Cellular Adaptation

Mechanical forces exert significant influence on cellular behavior, particularly in muscle development and regeneration. These forces can directly impact gene transcription related to myogenesis, activating transcription factors like MYOD1, which are crucial for processes such as myoblast fusion, myofiber alignment, and overall muscle tissue functionality [133]. Mechanical stimulation also modulates intracellular calcium levels through stretch-activated channels (SACs), further influencing gene transcription and promoting muscle differentiation [143]. In vitro models using materials like acrylamide gels with adjustable stiffness are often employed to study how mechanical and biochemical signals integrate, simulating the mechanical properties of the ECM [140]. Additionally, extrinsic mechanical loading, such as applying force via a four-point bending device, allows researchers to tune substrate stiffness, thereby directly impacting the behavior of cell-seeded substrates and enhancing their regenerative potential [49]. The ECM itself provides essential structural support and biochemical cues that guide stem cell fate, with the cytoskeleton playing a crucial role in integrating these signals to ensure proper cellular adaptation and function [131].

## 7. Cellular-Based Strategies for Regeneration

### 7.1. Preloading Scaffolds with Multiple Cell Types

Preloading scaffolds with a variety of cell types, particularly muscle stem cells (MuSCs), has shown significant promise in promoting muscle formation and achieving partial functional restoration in volumetric muscle loss (VML) injuries [144]. These cellular-based strategies typically involve the use of autologous cells to avoid immune rejection, which presents a significant advantage over allogeneic approaches. However, the use of autologous cells necessitates sophisticated manufacturing processes, making cellular-based strategies more complex compared to acellular approaches [90]. Additionally, co-culturing MuSCs with immune cells, such as CD3+ T-cells, has been found to support extensive MuSC expansion, enhancing their therapeutic potential for muscle regeneration [145]. By incorporating a tailored mix of cytokines that reflect the inflammatory milieu, these strategies can sustain MuSC expansion in vitro, further improving their effectiveness. Ref. [76] provides insights into how cytokines like IL-6 and TNF-alpha can be utilized to modulate the inflammatory environment, sustaining MuSC expansion and improving therapeutic outcomes. Ref. [79] highlights the importance of creating a supportive niche for muscle stem cells, which can be enhanced by a cytokine mix.

### 7.2. Pharmacological Interventions in Muscle Repair

Pharmacological interventions play a crucial role in enhancing muscle repair by modulating the cellular environment and supporting the function of both cellular and acellular therapies. Biomaterial scaffolds, for example, can be engineered to deliver growth factors like IGF-1 and FGF-2 directly to the injury site, stimulating satellite cell proliferation and improving muscle repair outcomes [146]. Additionally, pharmacological agents such as MMP inhibitors are employed to prevent excessive ECM degradation, which is vital for maintaining the structural integrity of biomaterial scaffolds [110]. Another approach involves using MSC-conditioned medium to precondition MuSCs before implantation, thereby improving their regenerative capabilities and enhancing overall muscle repair [147]. These pharmacological strategies are designed to modulate the cellular environment, thereby increasing the effectiveness of therapeutic interventions (Figure 6) [107].

### 7.3. Integration of Bioactive Molecules

Integrating bioactive molecules into scaffolds is essential for minimizing immune rejection, reducing inflammation, and promoting long-term tissue regeneration. Strategies such as using biocompatible materials, applying surface modifications, and incorporating anti-inflammatory agents into the scaffold help achieve these goals [139]. For instance, seeding decellularized ECM patches with MuSCs or other progenitor cells can enhance muscle repair by improving cell attachment and tissue integration [57]. Additionally, exosomes carrying growth factors like IGF-1 or FGF-2 have been shown to specifically promote MuSC proliferation and differentiation, while exosomes containing anti-inflammatory miRNAs help resolve inflammation, facilitating a conducive environment for tissue repair [99]. The Smad/AKT pathways, activated by follistatin-enriched exosomes from umbilical cord MSCs, also play a critical role in reducing fibrosis and enhancing muscle regeneration [148]. Moreover, modulating Wnt signaling through ASC-exosomes can alter macrophage phenotypes, further enhancing tissue repair through multiple signaling pathways [149].

### 7.4. Exosome and Immune Modulation

Exosomes derived from mesenchymal stem cells (MSCs) have emerged as a powerful tool for modulating the immune response and promoting MuSC proliferation and differentiation. These exosomes carry a bioactive cargo of proteins, lipids, and RNAs, which contribute to their therapeutic potential [150]. Exosomes that are specifically engineered to carry IGF-1 or FGF-2 have been demonstrated to enhance MuSC proliferation and differentiation, while those loaded with anti-inflammatory miRNAs help mitigate inflammation at the injury site [151]. Compared to traditional cell-based therapies, exosomes offer several advantages, including a lower likelihood of eliciting an immune response, scalability in production, and the ability to deliver targeted therapeutic cargos [152]. However, critical challenges remain, such as ensuring the targeted delivery of exosomes to injury sites, optimizing their therapeutic cargo, understanding their long-term effects, and standardizing production and characterization for clinical use [101]. Additionally, the WDR5-linked network associated with glutamine metabolism plays a role in promoting differentiation and supporting MuSC self-renewal by maintaining mitochondrial function, adding another layer of complexity and potential to exosome-based therapies (Table 3) [148].

**Table 3 bioengineering-11-01245-t003:** Enhancing MuSC youthfulness and proliferative potential: a summary of recent pharmacotherapy advances. This table summarizes recent advances in pharmacological and cellular strategies to enhance the youthfulness and proliferative potential of MuSCs.

Ref.	Molecular Pathway	Signaling Mechanism	MuSC State Affected	Outcome
[153]	PRMT inhibition	Type I PRMT inhibition	Proliferation	Enhances MuSC proliferation and muscle regeneration in Duchenne muscular dystrophy model
[154]	CXCL12/CXCR4 axis	MSC homing and early myogenesis	Early myogenesis	Enhances muscle repair through MSC homing, especially with intrarectal administration
[155]	MSC-conditioned medium	Anti-inflammatory and anti-fibrotic	New muscle fiber formation	Reduces inflammation and fibrosis, promotes new muscle fiber formation
[156]	Notch signaling	DLL1 bioprinting, MuSC maintenance	Maintenance and engraftment	Improves MuSC maintenance and engraftment in dystrophic muscles
[157]	Smad/AKT pathways	Follistatin-enriched exosomes	Fibrosis reduction and muscle regeneration	Reduces fibrosis and enhances muscle regeneration
[77]	Wnt signaling	ASC exosomes mediated macrophage phenotype modulation	Tissue repair	Enhances tissue repair through Wnt signaling and macrophage phenotype alteration
[158]	KDR signaling	Dystrophin complex, asymmetric division	Progenitor generation	Promotes muscle regeneration by enhancing progenitor generation
[159]	Histone demethylation	JMJD3-mediated H3K27 demethylation	Muscle repair	Activates MuSCs and enhances muscle repair through demethylation
[160]	Dkk3 signaling	Baf60c regulation in myofibers	Paracrine signaling	Supports regulated muscle regeneration through paracrine signaling
[8]	Mechanosensitive channels	Piezo1 activation	Morphology and regenerative capacity	Enhances regenerative capacity by influencing MuSC morphology in dystrophic conditions
[161]	Dual-specificity phosphatases (DUSP13/27)	MyoD-mediated proliferation	Proliferation and differentiation transition	Regulates MyoD-mediated proliferation and transition to differentiation
[162]	HIF2A inhibition	Satellite cell differentiation	Proliferation and differentiation	Boosts proliferation and enhances MuSC regeneration
[163]	Setd7 inhibition	(R)-PFI-2-mediated MuSC expansion	Regenerative capabilities post-transplantation	Maintains regenerative capabilities and supports MuSC expansion post-transplantation
[164]	Translational modulation	C10-mediated delay in differentiation	Stemness maintenance	Delays differentiation, improving yield of cultured MuSCs and maintaining stemness
[165]	PAX7:GFP sorting	Fibrin microfiber bundles, myotube formation	Muscle regeneration	Enhances myotube formation and muscle regeneration from human pluripotent stem cell-derived myogenic progenitors
[166]	Bioactive muscle patches	Concurrent administration with proteins and MuSCs	Muscle repair	Improves muscle repair in trauma scenarios
[157]	Transcriptome reprogramming	Epigenome changes	Post-transplantation regenerative capacity	Enhances regenerative capacity through transcriptome reprogramming
[167]	iPSC correction	Interspecies generation	Functional stem cell production	Produces functional stem cells, addressing therapeutic production and donor shortages
[168]	MSC exosome-mediated immune modulation	Macrophage polarization	Inflammation reduction, tissue repair	Reduces inflammation and enhances muscle regeneration through macrophage polarization
[169]	MSC exosome-mediated immune modulation	Macrophage polarization	Inflammation reduction, tissue repair	Enhances cartilage repair and reduces inflammation through macrophage polarization
[170]	Glutamine metabolism	WDR5-linked network	Differentiation and self-renewal	Promotes differentiation and supports self-renewal by maintaining mitochondrial function
[160]	Mitophagy	PINK1/Parkin pathway	Mitochondrial quality, ROS reduction	Improves mitochondrial quality, reduces ROS, and supports MuSC self-renewal
[164]	Lipid metabolism	LD turnover	Balanced differentiation and self-renewal	Ensures balanced fate decisions between LDLow and LDHigh MuSC differentiation
[161]	Pitx2 regulation	Myogenic precursors	Satellite cell proliferation and differentiation	Enhances satellite cell proliferation and differentiation in specific subpopulations
[8]	Primary cilia	Talpid3 (TA3) regulation	Regeneration and self-renewal	Enhances MuSC regeneration and self-renewal via Hedgehog signaling
[159]	Epigenetic regulation	SETDB1-mediated genome integrity	Genome integrity, aberrant activation prevention	Preserves genome integrity and prevents aberrant activation
[171]	Metabolic regulation	SIRT1 activation and caloric restriction	MuSC activation and function	Enhances MuSC activation and function through histone modifications
[172]	Nanotopographical cues	Engineered EVs, sequential administration	Proliferation, aged muscle repair	Enhances proliferation and aged muscle repair through engineered EVs

## 8. Advanced Analytical Techniques

### 8.1. Single-Cell RNA Sequencing (scRNA-Seq)

Single-cell RNA sequencing (scRNA-seq) has revolutionized our ability to analyze the cellular and molecular mechanisms underlying muscle repair. This advanced technique allows for the detailed examination of individual cells within the muscle tissue, providing insights into the dynamic changes occurring in muscle stem cells (MuSCs) and their surrounding niche during regeneration. By revealing distinct cell populations and their functional states, scRNA-seq enables researchers to map out the intricate interactions and regulatory networks that govern muscle repair processes [173]. Furthermore, combining scRNA-seq with spatial transcriptomics links molecular data with histological features, offering an enhanced understanding of tissue organization and the functional roles of different cell types during muscle regeneration [173].

### 8.2. Advanced Imaging Techniques

Advanced imaging techniques are crucial for providing spatial context to the molecular data obtained through techniques like scRNA-seq. Methods such as confocal microscopy and multiphoton microscopy allow for high-resolution, three-dimensional visualization of cell behavior, enabling real-time tracking of cellular and molecular events within muscle tissue [174]. Intravital microscopy and live cell imaging take this a step further by allowing for the observation of these processes in real time within live animals, offering invaluable insights into the dynamic nature of muscle regeneration [175]. These imaging techniques, when combined with molecular data, allow for a comprehensive understanding of how cells interact and function within the tissue during the repair process [175].

### 8.3. Combining Molecular Data with Histological Features

The integration of molecular data with histological features is becoming increasingly sophisticated, thanks to advances in single-cell proteomics and spatial profiling technologies. These approaches provide a comprehensive assessment of tissue composition and structure, which is essential for unraveling the complex interactions between different cell types during muscle regeneration [176]. For example, matrix elasticity is known to guide mesenchymal stem cells (MSCs) to differentiate according to the stiffness levels typical of their lineage-specific environments, which in turn affects their morphology, spreading, and the formation of cytoskeletal structures [177]. By combining these advanced molecular and imaging techniques, researchers are able to gain detailed insights into the cellular and molecular mechanisms at play, enabling the precise mapping of cellular interactions and regulatory networks during muscle repair [174].

## 9. Scaffold Types and Properties

### 9.1. Decellularized ECM Scaffolds

Decellularized extracellular matrix (ECM) scaffolds are highly regarded for their ability to provide a structural framework that facilitates host cell infiltration and tissue regeneration. These scaffolds are advantageous due to their scalability and cost-effectiveness, making them a popular choice in regenerative medicine [102]. Derived from donor muscle tissue, decellularized ECM scaffolds preserve critical structural and biochemical cues, which are essential for guiding host cell behavior during tissue regeneration [11]. A specific approach involves biologically derived scaffolds created using wet electrospinning, which are composites functionalized with ECM hydrogel. These scaffolds support human mesenchymal stem cells (hMSCs) and exhibit moderate biodegradability, high proangiogenic properties, and moderate sustainability, although they have low electrical properties [178]. Additionally, muscle-specific decellularized ECM scaffolds incorporated with IGF-1 have shown high sustainability and proangiogenic properties, supporting myoblast proliferation despite moderate biodegradability and low electrical properties [179].

### 9.2. Synthetic and Bioactive Muscle Patches

Synthetic and bioactive muscle patches are designed to provide structural support and deliver bioactive cues that promote muscle repair and integration with host tissue. These patches can be enhanced by seeding with MuSCs or other progenitor cells, which improves muscle repair by facilitating cell attachment and tissue integration [102]. For example, polycaprolactone scaffolds, which are fluorescent and fabricated using 3D printing, are conjugated with near-infrared (NIR) technology to support placental stem cells. These scaffolds are characterized by slow biodegradability and low proangiogenic and electrical properties, but they are highly cost-effective, though less sustainable [178]. Moreover, bioactive muscle patches, when combined with proteins and MuSCs, have demonstrated enhanced muscle repair capabilities, particularly in traumatic injury scenarios [180].

### 9.3. Composite Scaffolds and Their Functionalization

Composite scaffolds combine various materials to optimize their mechanical and biological properties for tissue engineering applications. For instance, alginate–silk composite scaffolds, created via 3D bioprinting, are designed for dual differentiation and support hMSCs. These scaffolds offer moderate biodegradability, proangiogenic, and electrical properties, along with high cost-effectiveness and moderate sustainability [181]. Another example is collagen/polypyrrole composite scaffolds, which are freeze-dried to achieve alignment and conductivity. These scaffolds, functionalized with polypyrrole, support myoblasts and are known for their slow biodegradability, moderate proangiogenic properties, and high electrical properties, alongside moderate sustainability [182]. Gelatin/chitooligosaccharide/DBM composite scaffolds, produced through lyophilization, are porous and support MSCs. They exhibit moderate biodegradability, proangiogenic properties, and cost-effectiveness, with low electrical properties [183]. Bioactive ceramics/silk fibroin composite scaffolds, created using 3D printing, support hMSCs and offer moderate biodegradability and proangiogenic properties, with low electrical properties and high cost-effectiveness (Figure 7) [184].

### 9.4. Three-Dimensional-Printed and Electrospun Scaffolds

Three-dimensional printing technology allows for the creation of highly customized scaffolds tailored to match specific injury sites, providing personalized solutions for muscle regeneration [181]. This process enables the incorporation of multiple cell types and bioactive molecules, resulting in complex tissue constructs that promote cell proliferation and integration with existing tissue [185]. Electrospinning is another key technique used to create nanofibrous scaffolds that closely mimic the ECM of native muscle tissue, offering a high surface area and porosity for optimal cell attachment and nutrient exchange. Ref. [80] discusses the role of ECM components in tissue regeneration, with electrospun scaffolds providing an ideal environment for muscle cells. Ref. [186] describes how engineered scaffolds, including electrospun ones, mimic the native ECM, supporting cell attachment and function. These electrospun scaffolds can be further functionalized with bioactive molecules to enhance cell adhesion and differentiation, with aligned fibers guiding regenerating muscle fibers effectively [182]. Additionally, innovations such as “Drydux Transduction” have been developed to increase activation efficiency through viral vectors during the transduction process, improving the functionality of these scaffolds in regenerative applications (Figure 8 and Table 4).

**Figure 8 bioengineering-11-01245-f008:**
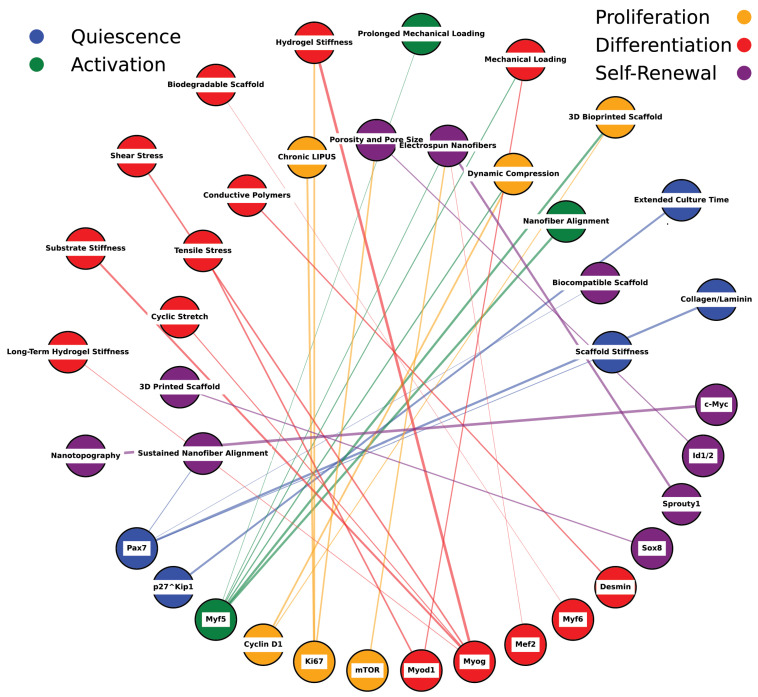
Network mapping of scaffold properties and muscle stem cell markers. This figure illustrates the estimated relationships between various scaffold properties (e.g., stiffness, alignment, porosity) and muscle stem cell markers, including Pax7, Myf5, Myod1, Myogenin, and others. The network displays interactions based on literature-derived estimates linking scaffold features to different stem cell fates such as quiescence, activation, proliferation, differentiation, and self-renewal. Scaffold properties, represented as nodes, interact through specific pathways with muscle stem cell markers and their regulatory networks. Color-coded nodes indicate stem cell fates influenced by scaffold design: blue represents quiescence, green denotes activation, yellow signifies proliferation, red indicates differentiation, and purple corresponds to self-renewal. The pathways connecting these nodes highlight the mechanistic role of scaffold properties in modulating muscle stem cell behavior and fate, reflecting how physical cues from scaffolds translate into cellular responses. This diagram emphasizes the complexity and interconnectedness of scaffold-induced regulation, illustrating their critical influence on stem cell signaling and tissue regeneration dynamics. The network is informed by selective literature reviews, as referenced in Table 4.

**Table 4 bioengineering-11-01245-t004:** Molecular pathways and signaling in MuSCs. The table categorizes the research by scaffold type, signal transduction mechanisms, cellular responses, and key molecules or proteins involved.

Ref.	Scaffold Type	Signal Transduction	Cellular Response	Key Molecules/Proteins
[187]	Nanotopography	FAK activation	Stem cell differentiation, ECM production	FAK, Cytoskeletal proteins
[188]	Meso-scale topological cues	Mechanotransduction	ECM production	FAK, ECM components
[189]	Aligned fibrous scaffolds	Integrin–FAK pathway	Muscle progenitor maturation	Integrin, FAK
[190]	Fibrous scaffolds	FAK activation	Skeletal regeneration, progenitor cell maturation	FAK
[191]	Electrospun fibers	Rho GTPase, FAK pathways	Tenogenic differentiation	Rho GTPase, FAK
[192]	Conductive nanofibers	MAPK/ERK pathway	Myotube formation	MAPK, ERK
[193]	Conductive nanofibers	MAPK/ERK pathway	Mechanotransduction, myotube formation	MAPK, ERK
[194]	Vitronectin, fibronectin, collagen	Integrin-FAK/Src pathway	Smooth muscle cell adhesion and proliferation	Integrin, FAK, Src
[195]	Vitronectin, fibronectin	Integrin-mediated mechanotransduction	Vascular smooth muscle cell behavior	Integrin, ECM components
[50]	Various ECM components	MAPK/ERK pathway	Myogenic differentiation	MAPK, ERK
[196]	Stiff scaffolds	RhoA/ROCK pathway	Cell behavior modulation, differentiation	RhoA, ROCK
[197]	Stiff scaffolds	RhoA/ROCK pathway	Response to mechanical cues, differentiation	RhoA, ROCK
[157]	Growth factors	JAK/STAT pathway	Satellite cell activation, tissue repair	JAK, STAT
[23]	Growth factors	JAK/STAT pathway	Satellite cell activation	JAK, STAT
[171]	Biochemical cues	Notch pathway	Satellite cell activation and differentiation	Notch
[77]	Biochemical cues	Notch pathway	Satellite cell quiescence, differentiation	Notch
[172]	Curved nanofiber networks	Actomyosin filament pathway	Osteogenic differentiation	Actin, Myosin
[198]	Graphene oxide substrates	Cytoskeletal protein reorganization	Cell migration, differentiation	Cytoskeletal proteins
[199]	Grooved patterns, chiral nematics	Myogenin, MyoD activation	Myotube formation	Myogenin, MyoD
[200]	Nanogratings	Extracellular vesicles (EVs) and myogenic proteins	Myogenic differentiation	EVs, Myogenic proteins
[201]	3D topographical features	PGE2, IL-6, MCP-1	Anti-inflammatory response	PGE2, IL-6, MCP-1
[202]	Mesoporous silica nanoparticles	PI3K/Akt pathway	Vascularization, drug delivery	PI3K, Akt
[203]	Shape memory polymers	MyoD, Myogenin signaling	Myogenic differentiation	MyoD, Myogenin
[204]	Motor amphiphiles	Membrane potential, calcium signaling	Controlled differentiation	Membrane potential, Calcium signaling
[160]	Electrospun PLCL scaffolds	Myogenic markers activation	Myogenesis	MyoD, Myogenin
[205]	Drydux scaffolds	Fluid flow optimization	Enhanced viral vector transduction efficiency	Viral vectors
[206]	Barium-doped calcium silicate	CaSR, AKT signaling	Osteogenic differentiation	CaSR, AKT
[207]	Degradable scaffolds	Wnt/β-catenin pathway	Tissue regeneration	Wnt, β-catenin
[97]	Shear stress applied scaffolds	Wnt signaling	Mesoderm differentiation	Wnt
[158]	Mechanical stimulation	MAPK pathway	Muscle adaptation	MAPK
[208]	Dynamic cellular projections	Rac-Rho GTPase pathway	Injury response, stem cell migration	Rac, Rho GTPase
[170]	Glutamine metabolism	WDR5-APC/C interaction	Stem cell differentiation	WDR5, APC/C

## 10. Integration of Emerging Technologies

### 10.1. Role of Biocompatible Materials in Scaffold Design

The selection of biocompatible and biodegradable materials is crucial for the successful clinical translation of scaffolds in muscle regeneration. These materials must ensure not only compatibility with host tissues but also degrade at a rate that supports tissue healing and integration without causing adverse reactions [209]. For instance, bioactive ceramics/silk fibroin composite scaffolds, created through 3D printing, support human mesenchymal stem cells (hMSCs) and offer a balance of moderate biodegradability and proangiogenic properties. These scaffolds are also cost-effective and moderately sustainable, although they possess low electrical properties, which may limit certain applications. Ref. [81] notes that while electrospun scaffolds are cost-effective, they may require modification to enhance electrical properties for specific applications. Ref. [77] emphasizes the need for improving scaffold properties to better support tissue engineering applications, particularly where electrical conductivity is important. Hybrid scaffolds, which combine hydrogels with electrospun fibers or 3D-printed frameworks, offer enhanced mechanical support while maintaining biological activity, making them ideal for optimizing tissue repair processes [121,210]. Similarly, decellularized ECM scaffolds, with their fibrous and porous structure, provide a supportive environment for human adipose-derived stem cells (hASCs) and exhibit high sustainability, despite moderate biodegradability and proangiogenic properties [211].

### 10.2. Functionalization for Enhanced Regeneration

Functionalization of scaffolds and therapeutic strategies plays a key role in enhancing muscle regeneration. Pharmacological interventions are often employed to modulate the cellular environment, making both cellular and acellular therapies more effective. Ref. [72] illustrates how pharmacological agents can modulate the cellular environment, enhancing the effectiveness of regenerative therapies. Ref. [74] discusses how pharmacological interventions can support tissue repair by modulating the inflammatory response and cellular activity. Exosomes carrying growth factors like IGF-1 or FGF-2 can specifically enhance MuSC proliferation and differentiation, while exosomes containing anti-inflammatory miRNAs help mitigate inflammation, creating a conducive environment for muscle repair [102]. Additionally, histone marks, particularly those influenced by pathways interconnected with MuSC activation, play a significant role in facilitating differentiation and muscle repair. Ref. [78] highlights the role of histone modifications in regulating gene expression during MuSC activation and differentiation, which is crucial for muscle repair. Histone demethylases such as JMJD3 are critical in this context, as they activate MuSCs by demethylating H3K27, thereby improving muscle regeneration outcomes [212]. Scaffold alignment also contributes to enhanced cellular proliferation by activating integrin-mediated signaling via the PI3K/Akt pathway, which is vital for effective tissue regeneration [213].

### 10.3. Nanotechnology and 3D Topographical Cues

Nanotechnology and 3D topographical cues have become integral to guiding cell behavior and improving tissue integration in regenerative medicine (Table 5). Surface features at the microscale and nanoscale influence cytoskeletal alignment and focal adhesion dynamics, which are essential for stem cell differentiation and effective tissue repair [214]. For example, curved nanofiber networks have been shown to promote osteogenic differentiation by engaging the actomyosin filament pathway, which involves both actin and myosin [213]. Nanogratings, on the other hand, promote myogenic differentiation through the action of extracellular vesicles (EVs) and the expression of myogenic proteins. Ref. [75] explores how nanostructures like nanogratings can influence cell behavior, promoting myogenic differentiation through EV signaling. Ref. [69] examines how extracellular vesicles carry signals that promote myogenic protein expression and differentiation, supported by nanogratings. Furthermore, 3D topographical features are crucial in regulating the crosstalk between mesenchymal stem cells (MSCs) and macrophages, leading to an anti-inflammatory response mediated by factors like PGE2, IL-6, and MCP-1 [214]. Nanotopographical cues, when delivered via engineered EVs, have shown potential in enhancing cell proliferation and aiding in the repair of aged muscle tissue, particularly through sequential administration strategies [215].

**Table 5 bioengineering-11-01245-t005:** Characterization of scaffold innovations for stem cell-based skeletal muscle tissue engineering.

Ref.	Scaffold Material	Source	Fabrication Technique	Structural Features	Cell Type Supported	Biodegradability	Proangiogenic Properties	Electrical Properties	Sustainability
[216]	Polycaprolactone (PCL)	Synthetic	3D Printing	Fluorescent features, NIR conjugation	Placental stem cells	Slow	Low	Low	Low
[217]	Polycaprolactone (PCL)	Synthetic	3D Printing	Structural stability	Placental stem cells	Slow	Low	Low	Low
[218]	Decellularized ECM	Biological	Decellularization	Spheroid-derived structures	Endothelial cells	Moderate	High	Low	Moderate
[219]	Decellularized ECM	Biological	3D Bioprinting	Spheroid-derived structures	Endothelial cells	Moderate	High	Low	Moderate
[220]	Cellulose	Plant	Decellularization	Grooved structural characteristics	C2C12, HSMCs cells	Slow	Low	Low	High
[221]	Cellulose	Plant	Decellularization	Grooved structural characteristics	C2C12, HSMCs cells	Slow	Low	Low	High
[222]	Collagen	Biological	Freeze-drying	Fibrillar structures	MSCs	Moderate	Low	Low	Moderate
[223]	Collagen	Biological	Freeze-drying	Fibrillar structures	MSCs	Moderate	Low	Low	Moderate
[224]	Polycaprolactone (PCL)/Gold	Synthetic	Melt Electrowriting	Hierarchical, anisotropic structure, Gold coating	H9c2 myoblasts	Slow	Moderate	High	Low
[225]	Polycaprolactone (PCL)/Polypyrrole	Synthetic	Melt Electrowriting	Hierarchical, anisotropic structure, Gold coating	H9c2 myoblasts	Slow	Moderate	High	Low
[226]	Gelatin/Chitooligosaccharide/DBM	Composite	Lyophilization	Porous structure	MSCs	Moderate	Moderate	Low	Moderate
[227]	Gelatin/Chitosan	Composite	Salt-leaching/Lyophilization	Porous structure	MSCs	Moderate	Moderate	Low	Moderate
[228]	Collagen/Polypyrrole	Composite	Freeze-drying	Aligned, conductive structural features	Myoblasts	Slow	Moderate	High	Moderate
[221]	Plant-derived Cellulose	Plant	Decellularization	Striated structures	C2C12 cells	Slow	Low	Low	High
[216]	Polycaprolactone (PCL)	Synthetic	3D Printing	Artificial ECM functionalization	hASCs	Slow	High	Low	Low
[229]	Alginate–Gelatin	Composite	3D Bioprinting	Hierarchical structural features	hMSCs	Moderate	Low	Low	Moderate
[230]	Alginate–Gelatin	Composite	3D Bioprinting	Hierarchical structural features	hMSCs	Moderate	Low	Low	Moderate
[179]	Decellularized ECM with IGF-1	Biological	Decellularization	Muscle-specific structures, IGF-1 incorporation	Myoblasts	Moderate	High	Low	High
[231]	Decellularized ECM with topographical cues	Biological	Wet Electrospinning	Composite ECM hydrogel structures	hMSCs	Moderate	High	Low	Moderate
[11]	Decellularized ECM	Biological	Wet Electrospinning	Composite ECM hydrogel structures	hMSCs	Moderate	High	Low	Moderate
[232]	Decellularized ECM	Biological	Decellularization	Fibrous, porous structures	hASCs	Moderate	Moderate	Low	High
[233]	Decellularized ECM	Biological	Infusion Bioreactor	Fibrous, porous structures	hASCs	Moderate	Moderate	Low	High
[150]	Bioactive Ceramics/Silk Fibroin	Composite	3D Printing	Composite structures	hMSCs	Moderate	Moderate	Low	Moderate
[228]	Bioactive Ceramics/Silk Fibroin	Composite	3D Printing	Composite structures	hMSCs	Moderate	Moderate	Low	Moderate

## 11. Future Directions

### 11.1. Hybrid Scaffold Systems

The development of hybrid biomimetic scaffolds that integrate ECM-mimicking hydrogels with nanofibrous architectures through advanced 3D bioprinting and electrospinning techniques represents a significant direction in replicating the anisotropic mechanical properties and biochemical cues of native muscle tissue. Future efforts should focus on engineering scaffolds with dynamic viscoelastic properties, utilizing stimuli-responsive materials that adapt stiffness and degradation rates in response to microenvironmental changes. This approach will ensure synchronized support throughout the various stages of tissue regeneration. Additionally, incorporating multi-modal bioactive molecules, including growth factors, cytokines, and epigenetic modulators, into scaffold matrices will allow for the spatiotemporal control of MuSC behavior, with a particular emphasis on developing controlled release systems that provide localized bioactivity.

### 11.2. Mechanotransduction Optimization

Investigating the role of mechanosensitive ion channels such as Piezo1 in cytoskeletal reorganization and nuclear mechano-signaling is essential to understanding their downstream effects on YAP/TAZ activation and gene transcription, particularly in the context of myogenesis. Designing substrates with tunable rigidity and topographical features will be critical in modulating focal adhesion dynamics, thereby enhancing the activation of pathways like RhoA/ROCK and FAK. These pathways are instrumental in promoting MuSC alignment, proliferation, and differentiation. Moreover, exploring the crosstalk between mechanotransduction and metabolic reprogramming in MuSCs, especially the impact of actomyosin contractility on glycolysis and oxidative phosphorylation, will provide insights into sustaining cellular energetics during muscle repair.

### 11.3. Epigenetic Scaffold Integration

The integration of epigenetic modulators within scaffolds presents an innovative strategy for regulating MuSC behavior during muscle regeneration. Future scaffolds should be designed to incorporate histone deacetylase inhibitors (HDACi) or DNA methyltransferase inhibitors (DNMTi) to modulate chromatin accessibility, either maintaining MuSC quiescence or promoting their activation in response to injury. Additionally, incorporating non-coding RNAs such as miR-1 and lncRNAs within scaffold matrices could enhance post-transcriptional gene regulation and chromatin remodeling, improving the precision of MuSC fate determination. Employing CRISPR-dCas9-based epigenetic editing within scaffold environments offers the potential to precisely target myogenic regulatory regions, such as those controlling MyoD and Myf5, to modulate key transcription factors critical for MuSC differentiation and self-renewal.

### 11.4. Scalable Differentiation Protocols

Optimizing bioreactor-based culture systems is imperative for the large-scale production of iPSC-derived myogenic progenitors, with a focus on maintaining Pax7 expression and minimizing spontaneous differentiation during expansion phases. Integrating Wnt and FGF signaling modulators into differentiation protocols will be essential for enhancing the yield and purity of myogenic progenitors while minimizing the risk of teratoma formation in transplantation settings. Further exploration of 3D spheroid culture systems is needed, as these systems more accurately replicate the in vivo niche of myogenic progenitors, promoting cell–cell interactions and ECM deposition to improve the functional integration of transplanted cells.

### 11.5. Exosome Engineering for Targeted Delivery

The engineering of exosomes for targeted delivery to injured muscle tissue is a promising avenue for enhancing therapeutic outcomes. Surface modifications, such as PEGylation and antibody conjugation, should be explored to improve tissue-specific homing and reduce off-target effects. Utilizing CRISPR-based gene editing to modify exosome-producing cells will enable the production of exosomes enriched with myogenic miRNAs (e.g., miR-133) and pro-regenerative proteins (e.g., IGF-1, FGF-2). Developing multilayered exosome formulations, incorporating lipid nanoparticles or hydrogel carriers, will be critical for achieving sustained release and enhanced bioavailability at injury sites, with a focus on increasing the in vivo half-life of exosomal cargo.

### 11.6. Targeted Immune Modulation

Exosome-based therapies specifically engineered to modulate macrophage polarization, particularly the transition from M1 to M2 phenotypes, hold potential for reducing fibrosis and promoting a pro-regenerative microenvironment. These therapies should focus on exosome cargo enriched with anti-inflammatory miRNAs and TGF-β. Additionally, the localized delivery of senolytic agents, such as Dasatinib and Quercetin, within scaffolds offers a targeted approach to selectively eliminate senescent cells in aged muscle, thereby minimizing systemic side effects and enhancing the regenerative potential of MuSCs. The development of biodegradable immunomodulatory scaffolds that release cytokines such as IL-4, IL-10, or Tregs-inducing factors in a controlled manner will be crucial for orchestrating the spatial and temporal aspects of the immune response during muscle regeneration.

### 11.7. Real-Time Monitoring and Analysis

Future research should emphasize the use of intravital microscopy combined with fluorescent reporter systems to dynamically track MuSC activation, migration, and differentiation in real-time during tissue regeneration. This will provide critical insights into cellular behavior in vivo. Integrating single-cell RNA sequencing (scRNA-seq) with spatial transcriptomics will enable the mapping of molecular heterogeneity within MuSC populations at various stages of muscle repair, focusing on identifying key regulatory networks and cellular interactions. Additionally, applying 3D spatial profiling technologies to analyze ECM composition and cellular architecture within regenerating muscle tissue will be essential for correlating microenvironmental changes with regenerative outcomes and scaffold performance.

## 12. Conclusions

Muscle stem cells (MuSCs) are central to the process of skeletal muscle regeneration, with their activity intricately regulated by mechanotransduction pathways such as YAP/TAZ, Rho/ROCK, and PI3K/Akt. These signaling cascades, influenced by the biophysical properties of the extracellular matrix (ECM), guide MuSC fate through integrin-mediated focal adhesion dynamics and cytoskeletal reorganization. The strategic incorporation of bioactive molecules within engineered scaffolds offers a powerful tool for modulating these intracellular pathways, with growth factors like IGF-1 and FGF-2 playing a critical role in sustaining MuSC proliferation, differentiation, and myotube formation. This biomimetic approach ensures that scaffolds maintain their bioactivity in alignment with the temporal requirements of muscle tissue regeneration.

The use of pharmacological agents, particularly senolytics such as dasatinib and navitoclax, has shown promise in counteracting the adverse effects of cellular senescence within the aging MuSC niche. By selectively inducing apoptosis in senescent cells, these agents help preserve the regenerative capacity of MuSCs, contributing to tissue homeostasis. When combined with scaffold-based delivery systems, these drugs not only enhance their bioavailability but also maximize their therapeutic efficacy.

Technological advancements, including single-cell RNA sequencing (scRNA-seq) and high-resolution imaging modalities like multiphoton microscopy and spatial transcriptomics, have provided unprecedented insights into the cellular heterogeneity and molecular dynamics of regenerating muscle. These tools have enabled the identification of distinct MuSC subpopulations and the transcriptional networks that govern their activation and differentiation, offering a detailed, single-cell resolution understanding of muscle repair mechanisms.

The integration of these cutting-edge imaging techniques and molecular profiling tools has significantly advanced the field of muscle regeneration research. By unraveling the complex interactions between MuSCs, their niche, and the ECM, researchers have identified key regulatory nodes and epigenetic modifications that control MuSC behavior. The fusion of these insights with bioengineered scaffold technologies has optimized regenerative strategies, enhancing the potential to restore functional muscle architecture and paving the way for more effective tissue engineering approaches in clinical applications.

## Figures and Tables

**Figure 1 bioengineering-11-01245-f001:**
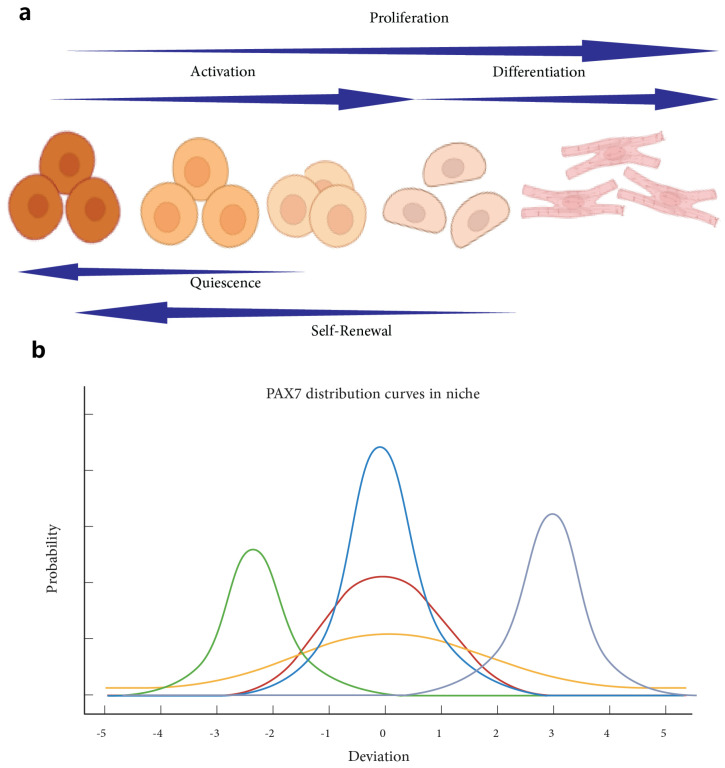
Understanding muscle stem cell states and PAX7 distribution. Understanding Muscle Stem Cell States: Quiescence, Activation, Proliferation, Differentiation, and Self-Renewal. Panel (**a**) illustrates the dynamic transition of muscle stem cells through various states, starting from quiescence, progressing through activation and proliferation, leading to differentiation, and the potential for self-renewal [29]. The schematic emphasizes the directional flow and sequential nature of these states, with each phase contributing to muscle regeneration and maintenance. Panel (**b**) presents PAX7 distribution curves within the niche, highlighting the variability and shifts in PAX7 expression levels during different states. The distribution curves illustrate how PAX7 expression peaks and shifts as muscle stem cells transition between quiescence, activation, and other states, reflecting their changing roles and regulatory mechanisms within the cellular environment [35]. Together, these panels provide a comprehensive overview of muscle stem cell state transitions and the critical role of PAX7 as a marker modulating these processes.

**Figure 2 bioengineering-11-01245-f002:**
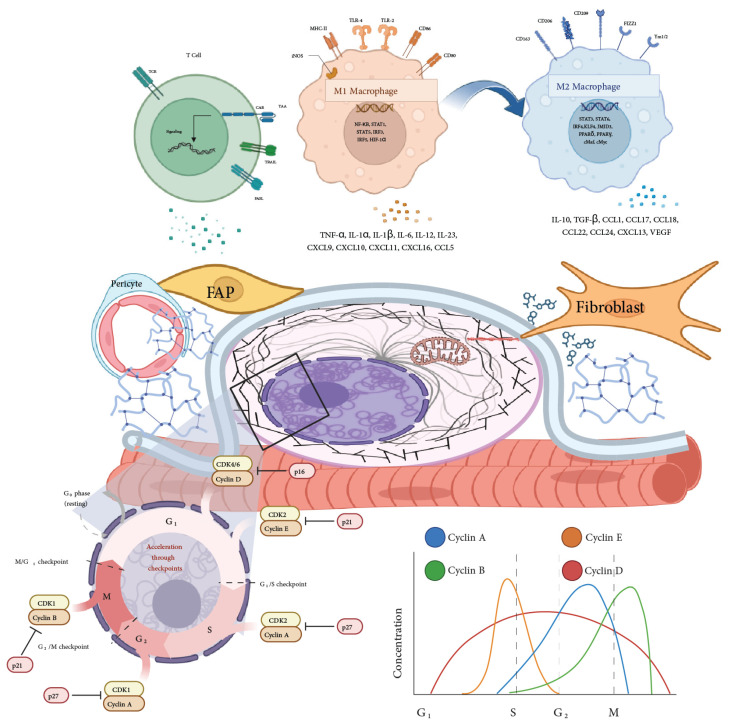
Cellular interactions and regulatory mechanisms in the muscle stem cell niche. This figure depicts the complex interplay among various cellular components of the muscle stem cell niche, including fibro-adipogenic progenitors (FAPs), pericytes, fibroblasts, and macrophages (M1 and M2 phenotypes). The top section illustrates the role of macrophages, showing the transition from pro-inflammatory M1 macrophages, which secrete cytokines like TNF-α and IL-1β, to anti-inflammatory M2 macrophages involved in tissue repair and regeneration, producing factors such as IL-10 and TGF-β. Adjacent panels highlight interactions with T cells and their influence on the local inflammatory response. The central portion emphasizes cellular and extracellular matrix (ECM) interactions mediated by FAPs, pericytes, and fibroblasts, influencing the structural and regenerative dynamics of the muscle niche. The figure also outlines cell cycle regulation, detailing checkpoints and the role of cyclins (Cyclin A, B, D, E) and cyclin-dependent kinases (CDKs) in governing progression through G1, S, G2, and M phases. The accompanying graph on cyclin and CDK concentration during the cell cycle phases provides a visual representation of how cyclin–CDK complexes fluctuate, coordinating MuSC proliferation and differentiation [47]. Specific cyclin–CDK complexes’ elevated levels promote cell cycle progression, while their inhibition leads to cell cycle arrest, thus tightly regulating MuSC proliferation within the niche [48]. Together, these elements underscore the regulatory pathways and cellular interactions that maintain muscle homeostasis and regeneration within the niche.

**Figure 3 bioengineering-11-01245-f003:**
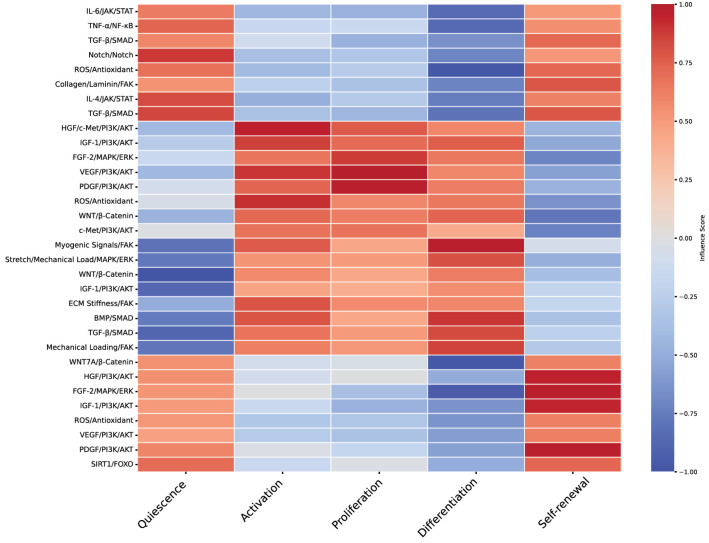
Heatmap of molecular pathways influencing muscle stem cell fates. This heatmap provides a visualization of the correlation between various molecular pathways and muscle stem cell fates, based on a qualitative exploratory review of relevant literature. The pathways considered include IL-6/JAK/STAT, TNF-α/NF-κB, TGF-β/SMAD, Notch, ROS/antioxidant mechanisms, and others. Muscle stem cell fates—quiescence, activation, proliferation, differentiation, and self-renewal—are represented along the *x*-axis, while molecular pathways are listed on the *y*-axis. The color gradient reflects the influence score of each pathway, with red hues indicating a positive correlation and blue hues representing a negative correlation. This visualization highlights how different pathways modulate specific cellular states, providing insight into the complex regulatory networks governing muscle stem cell behavior. The heatmap underscores the critical interplay of signaling cascades that dictate stem cell responses and tissue regeneration, as referenced in Table 1.

**Figure 4 bioengineering-11-01245-f004:**
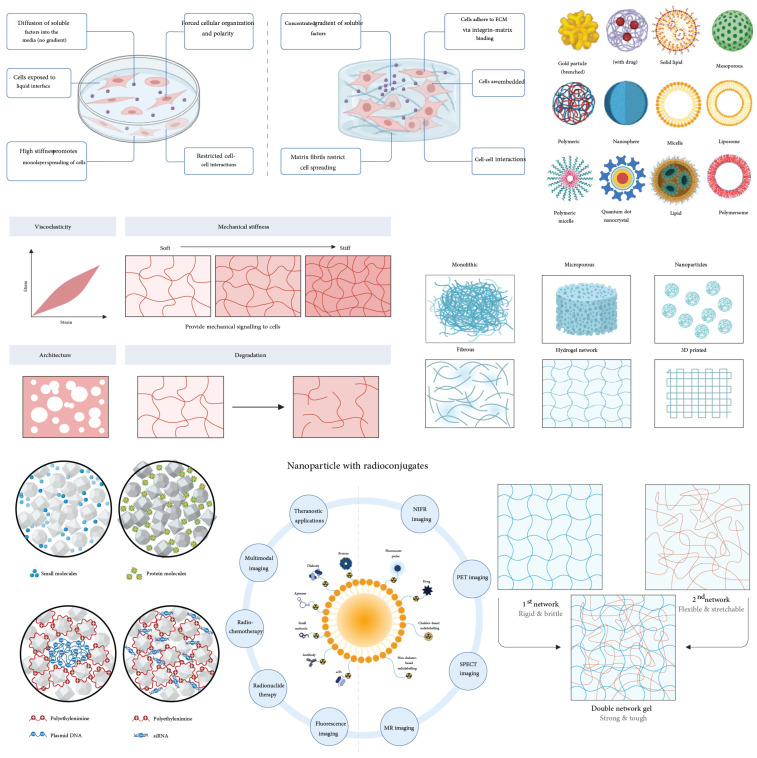
Scaffold characteristics. The figure provides an overview of critical scaffold properties that influence muscle stem cell function and tissue engineering outcomes. The properties are categorized into structural and functional aspects, including vascularity, architecture, pore size, degradation, mechanical properties, and nano-specific modifications. The upper left panels illustrate how cells interact with scaffolds, emphasizing effects on cellular organization, polarity, and matrix binding, driven by diffusion gradients and cell–extracellular matrix (ECM) interactions. The central portion of the figure demonstrates mechanical properties, ranging from viscoelasticity (depicting a strain-stress curve) to stiffness levels (soft to stiff matrices) that provide mechanical cues to cells, critical for their behavior and differentiation. Adjacent panels depict architectural considerations such as pore size and overall scaffold structure, influencing cell infiltration, tissue integration, and mass transport. Degradation profiles are shown, emphasizing scaffold dissolution kinetics that match tissue regeneration rates. On the right side, nano-specific modifications, including particles (solid lipid, polymeric vesicles, quantum dots, etc.), highlight applications such as drug delivery and enhanced cellular interactions. The bottom panels explore complex architectures, like double-network gels that combine rigid and flexible networks, enhancing mechanical resilience and adaptability in dynamic tissue environments. The integration of nanoparticles with radioconjugates illustrates potential therapeutic and imaging applications in regenerative medicine and diagnostics. The progressive narrative of the figure emphasizes the interplay of scaffold properties with biological, mechanical, and nanoscale features for optimizing tissue regeneration outcomes.

**Figure 5 bioengineering-11-01245-f005:**
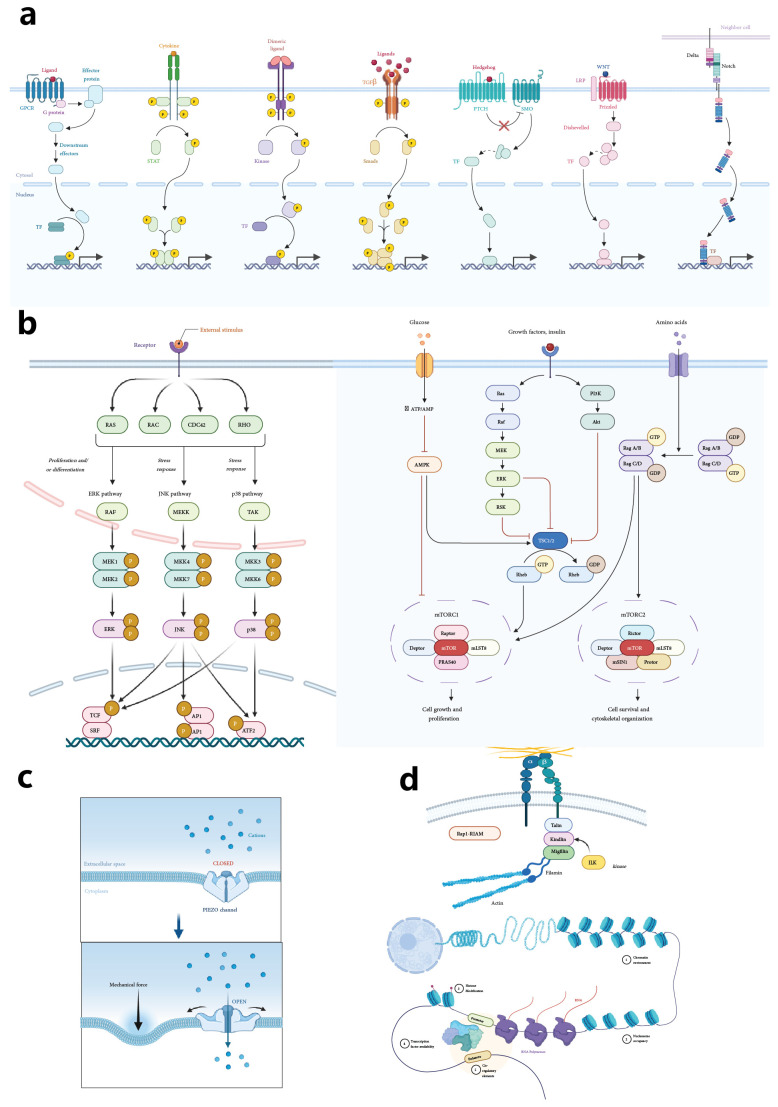
Mechanisms and pathways in muscle stem cell regulation. This figure encompasses multiple schematic representations that outline critical regulatory mechanisms influencing muscle stem cell behavior and fate. Panel (**a**) illustrates major signaling pathways, including Wnt, BMP, TGF-β, Notch, and PI3K/AKT pathways, emphasizing the journey from receptor activation at the cell membrane to downstream nuclear transcription factors that modulate gene expression, impacting muscle stem cell states and responses. Panel (**b**) provides detailed pathway diagrams, showcasing interactions among key regulatory molecules in muscle stem cell signaling. This includes cross-talk between different pathways that guide cell proliferation, differentiation, quiescence, and self-renewal, depicting an integrated regulatory network. Panel (**c**) highlights the role of mechanical forces, such as shear stress and tensile stress, in triggering cellular mechanotransduction processes, with a specific focus on mechanical receptors and their influence on intracellular pathways. Panel (**d**) illustrates epigenetic regulatory mechanisms, including DNA methylation, histone modification, and non-coding RNAs, which fine-tune muscle stem cell states by modifying chromatin structure and gene expression, contributing to the maintenance of stem cell quiescence, activation, or differentiation. Together, these panels underscore the complexity and integration of biochemical, mechanical, and epigenetic influences on muscle stem cell regulation.

**Figure 6 bioengineering-11-01245-f006:**
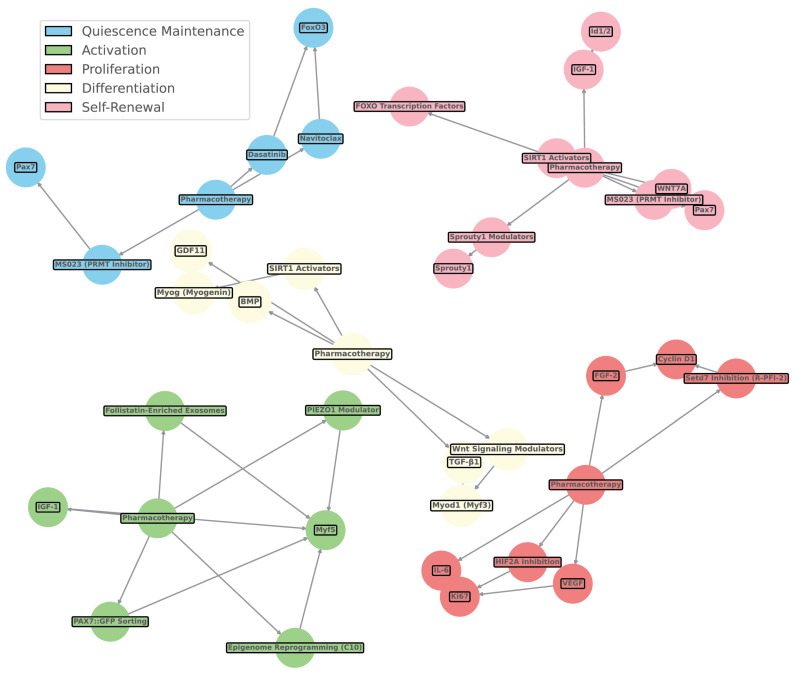
Network diagram of pharmacotherapy and muscle stem cell pathways. This figure visualizes the integration of pharmacotherapy interventions, such as growth factors, small molecules, and other modulators, with muscle stem cell pathways and markers. Nodes represent key elements of pharmacotherapy and stem cell signaling, while edges illustrate their interactions based on a qualitative exploratory review of the relevant literature. The color-coded nodes indicate distinct muscle stem cell states influenced by pharmacotherapy: blue for quiescence maintenance, green for activation, yellow for proliferation, red for differentiation, and purple for self-renewal. The network emphasizes the complex, multi-faceted approach to muscle regeneration, showcasing how targeted pharmacotherapy influences specific pathways, such as Wnt signaling modulators, SIRT1 activators, BMP signaling, and more. This comprehensive view of pharmacological modulation highlights the interplay between small molecules, growth factors, and muscle stem cell regulatory networks, facilitating targeted therapeutic strategies for muscle repair and regeneration, as referenced in Table 3.

**Figure 7 bioengineering-11-01245-f007:**
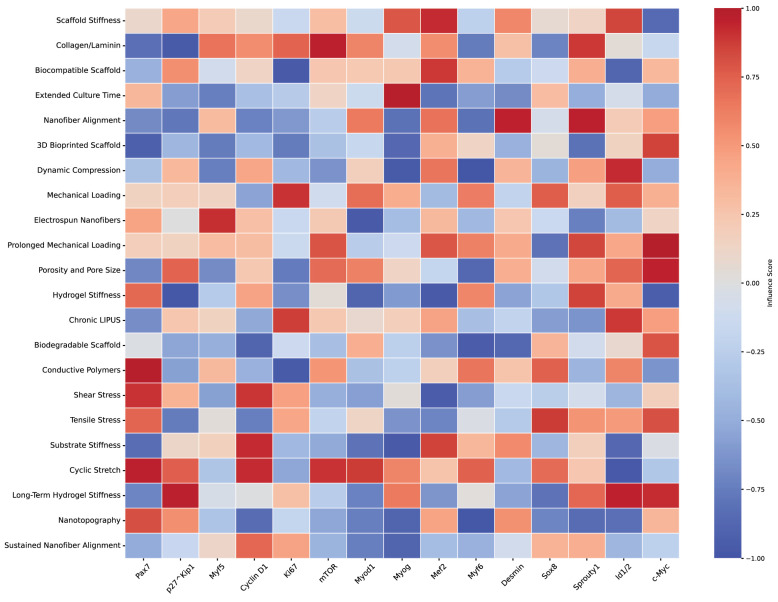
Heatmap of scaffold characteristics impact on muscle stem cell markers. This heatmap visually represents the estimated impact of various scaffold characteristics on muscle stem cell markers, based on a non-systematic literature review. The scaffold properties examined include stiffness, porosity, mechanical loading, alignment, and others. Muscle stem cell markers (e.g., Pax7, Myf5, Myod1, Myogenin) are displayed along the *x*-axis, while scaffold characteristics are on the *y*-axis. Color intensity and direction indicate the correlation strength, with red hues representing positive correlations and blue hues showing negative correlations. These interactions reflect the influence of scaffold design on stem cell behaviors such as quiescence, activation, proliferation, differentiation, and self-renewal. The heatmap provides insights into how scaffold properties can be strategically modulated to optimize muscle regeneration and stem cell function, summarizing findings based on references listed in Table 4.

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
