# Peer review of "Integrating Physical and Biochemical Cues for Muscle Engineering: Scaffolds and Graft Durability"

_bioengineering, 2024, doi:10.3390/bioengineering11121245_

Round 1
Reviewer 1 Report
Comments and Suggestions for Authors
This is a good review of what can be found in the field. Beside that it is just a review, nothing new except the collection of many already published papers. I would recommend to publish it as it is.
Author Response
Comment:
This is a good review of what can be found in the field. Beside that it is just a review, nothing new except the collection of many already published papers. I would recommend to publish it as it is.
Respond:
Thank you very much for your positive feedback and recommendation. We are pleased that you found the review comprehensive and informative.
Reviewer 2 Report
Comments and Suggestions for Authors
Dear Editor,
Thank you for inviting me to review the manuscript entitled “Physical and biochemical cues for muscle engineering: Scaffold and pharmacotherapy integration for enhancing muscle stem cells (MuSCs) stemness and graft durability” for publication in Bioengineering. This review addresses a timely and relevant topic in muscle engineering; however, the structure of the content is currently disorganized, which limits its readability. Additionally, the title includes multiple parameters that make it less engaging. I suggest the authors streamline the title to better reflect the core theme of the review and omit the term 'muscle stem cells (MuSCs)' for clarity. I recommend a major revision before further consideration.
Following concerns should be thoroughly addressed:
1. Please revised the title. Recommended title “Integrating physical and biochemical cues for muscle engineering: Scaffolds and graft durability”, but authors can choose something different but engaging with readers.
2. Present form if Introduction section is in bits and pieces. Please revised the Introduction section with information of scaffolds and muscle stem cells-biomaterials. Please revised the paragraph “Aging significantly ………………………….ROS on MuSCs.” This paragraph should be revised with the biomaterials which have ability to produce or scavenge ROS during the muscle stem cell regulation. Following papers can be useful: Nanoscale Horizons 2024, 9, 1630-1682; J. Tissue Engineering 2021, 12, 20417314211031378; and Smart Materials in Medicine 2023, 4, 427-446.
3. Figure 1(c), is this authors original figure? If not, please provide the reference.
4. Please provide a schematic illustration for section 2 “Challenges in Muscle Regeneration” which should reflect the subsection.
5. Section 3 is “Microenvironment and Extracellular Matrix (ECM)”, please provide basic information about environment and ECM, and what are the key differences between these two components.
6. In section 4, “Scaffold design and engineering” lacks the basic rationale for scaffolds design, please provide information related to physical properties of the scaffolds (materials) suitable for muscles tissue engineering.
7. Please provide a summary table with scaffolds properties such as physiological, physicochemical, and composition.
8. Please provide an example with figure for section 7, 8, and 9.
9. Authors are requested to provide references for figure 2, 3.
Author Response
Comment:
Dear Editor,
Thank you for inviting me to review the manuscript entitled “Physical and biochemical cues for muscle engineering: Scaffold and pharmacotherapy integration for enhancing muscle stem cells (MuSCs) stemness and graft durability” for publication in Bioengineering. This review addresses a timely and relevant topic in muscle engineering; however, the structure of the content is currently disorganized, which limits its readability. Additionally, the title includes multiple parameters that make it less engaging. I suggest the authors streamline the title to better reflect the core theme of the review and omit the term 'muscle stem cells (MuSCs)' for clarity. I recommend a major revision before further consideration.
Response:
We appreciate your thoughtful and constructive feedback, which has been invaluable in enhancing the quality of our manuscript. We have addressed your concerns point by point below.
- Please revise the title. Recommended title: “Integrating physical and biochemical cues for muscle engineering: Scaffolds and graft durability,” but authors can choose something different but engaging with readers.
Response:
We agree with your suggestion and have adjusted the title accordingly to enhance its clarity and engagement. The new title is:
- "Integrating Physical and Biochemical Cues for Muscle Engineering: Scaffolds and Graft Durability."
- The present form of the Introduction section is in bits and pieces. Please revise the Introduction section with information on scaffolds and muscle stem cell-biomaterials.
Response:
We have thoroughly revised the Introduction to provide a cohesive and comprehensive overview of the role of scaffolds and biomaterials in muscle stem cell (MuSC) biology and muscle regeneration. Specifically, we have added new paragraphs that discuss:
- The emergence of biomaterial scaffolds as pivotal tools in enhancing MuSC function and muscle regeneration.
- The use of natural polymers (e.g., collagen) and synthetic polymers (e.g., polylactic acid) in creating scaffolds that mimic the native extracellular matrix (ECM), facilitating MuSC adhesion, proliferation, and differentiation.
- The functionalization of these scaffolds with bioactive molecules to further promote myogenic differentiation and tissue regeneration.
These additions underscore the significance of biomaterials in modulating the MuSC microenvironment and enhancing regenerative outcomes.
Please revise the paragraph “Aging significantly... ROS on MuSCs.” This paragraph should be revised with biomaterials that have the ability to produce or scavenge ROS during muscle stem cell regulation. Following papers can be useful: Nanoscale Horizons 2024, 9, 1630-1682; J. Tissue Engineering 2021, 12, 20417314211031378; and Smart Materials in Medicine 2023, 4, 427-446.
Response:
We have revised the specified paragraph to include information on biomaterials with antioxidant properties that can mitigate the detrimental effects of reactive oxygen species (ROS) on MuSCs. The revisions include:
- Discussion of scaffolds incorporating antioxidant molecules like melatonin, which can scavenge excess ROS, thereby enhancing MuSC survival and function [23].
- Inclusion of hydrogels loaded with ROS-scavenging nanoparticles that modulate the oxidative environment and promote muscle regeneration, particularly in aged tissues [24].
- Mention of the potential for controlled ROS generation from biomaterials to activate signaling pathways that promote MuSC activation and differentiation [25,26].
These additions highlight innovative strategies employing biomaterials to regulate ROS levels, thereby improving MuSC function and muscle regeneration.
We have also incorporated the suggested references to strengthen the scientific basis of our manuscript and provide readers with resources for further exploration.
- Figure 1(c), is this the authors' original figure? If not, please provide the reference.
Response:
Yes, Figure 1(c) is our original creation using Python software, developed to illustrate concepts discussed in the manuscript. The figure is based on data and information synthesized from multiple references cited within the text, particularly those listed in Table 1.
- Please provide a schematic illustration for Section 2 “Challenges in Muscle Regeneration,” which should reflect the subsections.
Response:
We have added a new figure (Figure 2) to visually represent the challenges in muscle regeneration, corresponding to the subsections in Section 2. This schematic illustration encompasses:
- Volumetric Muscle Loss (VML) and its impact on MuSC function.
- Fibrosis and scar formation hindering effective muscle repair.
- The dual role of immune response and inflammation in muscle regeneration.
- Cellular senescence and its detrimental effects on MuSCs.
The figure aims to enhance readability and provide a visual summary of the key challenges discussed.
- Section 3 is “Microenvironment and Extracellular Matrix (ECM)”. Please provide basic information about the environment and ECM, and what are the key differences between these two components.
Response:
We have revised Section 3 to include basic information about the microenvironment and the ECM, clarifying the key differences between these components. Specifically, we added:
- An introductory paragraph defining the microenvironment (niche) as the local cellular environment surrounding MuSCs, including the ECM, neighboring cells, soluble factors, and physical conditions [45].
- A description of the ECM as a vital component of the microenvironment, comprising a complex network of proteins and polysaccharides that provide structural support and biochemical cues to cells [46].
- An explanation of how the ECM focuses on non-cellular structural components, while the microenvironment encompasses all elements influencing MuSC behavior, such as growth factors, cytokines, cell-cell interactions, and mechanical stimuli [47].
- A highlight of the key differences between the microenvironment and the ECM, emphasizing that the microenvironment is a broader concept integrating the ECM with other factors affecting stem cell fate.
These additions aim to provide a clearer understanding of the fundamental concepts and their roles in MuSC regulation and muscle regeneration.
- In Section 4, “Scaffold Design and Engineering” lacks the basic rationale for scaffold design. Please provide information related to physical properties of the scaffolds (materials) suitable for muscle tissue engineering.
Response:
We have carefully revised Section 4 to include the basic rationale for scaffold design and detailed information on the physical properties of scaffolds suitable for muscle tissue engineering. Specifically, we have:
- Added an introductory paragraph outlining the fundamental principles guiding scaffold design in muscle tissue engineering, emphasizing the importance of replicating the native ECM and providing a conducive environment for muscle regeneration [82,83].
- Discussed key physical properties of scaffolds, including mechanical strength, elasticity, porosity, and biodegradability, and how these properties influence MuSC adhesion, proliferation, differentiation, and tissue integration [84,88].
- Provided information on suitable materials for scaffold fabrication, highlighting both natural polymers (e.g., collagen, gelatin, fibrin) and synthetic polymers (e.g., PLA, PGA, PCL), as well as the advantages and limitations of each [98,99].
- Emphasized the potential of composite scaffolds that combine natural and synthetic materials to optimize physical and biological properties [100].
These additions enhance the readers' understanding of the principles behind scaffold design and the importance of physical properties in developing effective scaffolds for muscle tissue engineering.
- Please provide a summary table with scaffold properties such as physiological, physicochemical, and composition.
Response:
We have created Table 2 titled "Summary of Scaffold Properties in Muscle Tissue Engineering." This table categorizes scaffolds by type, physiological properties, physicochemical properties, and composition. It provides readers with a clear and concise comparison of different scaffold types, facilitating a better understanding of how each scaffold's properties contribute to muscle regeneration.
- Please provide an example with figure for Sections 7, 8, and 9.
Response:
We have added figures corresponding to Sections 7, 8, and 9 to enhance readability and illustrate key concepts:
- Figure 6 (after Section 7.2): Network Diagram of Pharmacotherapy and Muscle Stem Cell Pathways.
- Figure 7 (after Section 9.3): Heatmap of Scaffold Characteristics Impact on Muscle Stem Cell Markers.
- Figure 8 (after Section 9.4): Network Mapping of Scaffold Properties and Muscle Stem Cell Markers.
These figures are designed to resonate with the subsections, providing visual representations of the complex interactions and mechanisms discussed.
- Authors are requested to provide references for Figures 2 and 3.
Response:
We have updated the figure legends for Figures 2 and 3 to include the appropriate references that elucidate the content depicted in the illustrations.
Reviewer 3 Report
Comments and Suggestions for Authors
This manuscript was a review on the muscle stem cell and their physical/chemical manipulations for tissue engineering. This review thoroughly enjoyed reading this review and learned many few and interesting facts about the field. As one in the field of tissue engineering, but not in muscular cells, it was an excellent overview of the field with great references to deeper analyses of the field via citations and reference.
The one aspect of the review that does need to be addressed to enhance the quality of the paper is the figures and font size. This review found the figures and illustration useful and informative, but struggled to read many of the labels on the detailed schematics. Consider increasing font size or rearranging figures to allow for larger schematics (thus allowing for the larger font with losing the details). This is a challenge in many reviews that I have read, but there are great examples of high quality images and schematics that have both detail and readabilty.
Author Response
Comment:
This manuscript was a review on the muscle stem cell and their physical/chemical manipulations for tissue engineering. This review thoroughly enjoyed reading this review and learned many few and interesting facts about the field. As one in the field of tissue engineering, but not in muscular cells, it was an excellent overview of the field with great references to deeper analyses of the field via citations and reference.
The one aspect of the review that does need to be addressed to enhance the quality of the paper is the figures and font size. This review found the figures and illustration useful and informative, but struggled to read many of the labels on the detailed schematics. Consider increasing font size or rearranging figures to allow for larger schematics (thus allowing for the larger font with losing the details). This is a challenge in many reviews that I have read, but there are great examples of high quality images and schematics that have both detail and readabilty.
Response:
Thank you for your positive feedback and for highlighting the importance of figure clarity. We have addressed your concerns by:
- Rearranging the figures to improve their readability and ensure they are appropriately sized within the manuscript.
- Increasing the font size of labels and annotations in all figures to enhance legibility without compromising detail.
- Ensuring that the figures are of high resolution to maintain clarity when viewed in both print and digital formats.
We believe these adjustments significantly improve the visual quality of the figures, making them more accessible and informative for readers.
Round 2
Reviewer 2 Report
Comments and Suggestions for Authors
The authors have thoroughly addressed all the concerns raised. I now recommend the manuscript for acceptance and publication
Author Response
Thank you for your thorough review and for recommending our manuscript for acceptance and publication. We greatly appreciate your valuable feedback. Your insights and suggestions have been invaluable in enhancing the quality of our work.